
# Class imbalance techniques for high energy physics

**Christopher W. Murphy**⋆

Insight Data Science, San Francisco, CA 94107, USA

⋆ chrismurphybnl@gmail.com

## Abstract

A common problem in a high energy physics experiment is extracting a signal from a much larger background. Posed as a classification task, there is said to be an imbalance in the number of samples belonging to the signal class versus the number of samples from the background class. In this work we provide a brief overview of class imbalance techniques in a high energy physics setting. Two case studies are presented: (1) the measurement of the longitudinal polarization fraction in same-sign $WW$ scattering, and (2) the decay of the Higgs boson to charm-quark pairs.

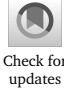
# 1 Overview

The Large Hadron Collider (LHC) has been an incredibly successful experiment. To date it has discovered the Higgs boson, and measured hundreds, if not thousands, of other processes to be consistent with the predictions of the Standard Model (SM) [1]. A common problem in making these measurements is extracting a signal from a much larger background. Occasionally in this situation there is a single feature that is powerful enough to discriminate the signal from the large background. An example of this the Higgs boson decaying to two photons where the invariant mass of the photon pair is the discriminating observable [2, 3]. More often however a multi-variate analysis of many features needs to be performed. Machine learning (ML) and deep learning (DL) are well suited for such tasks. Therefore it is not surprising that ML and DL have become, and will likely continue to be, an important part of the success of the LHC program. See Refs. [4–7] for some recent reviews.

If one treats the extraction of a signal from a much larger background as a classification problem there is an imbalance in the number of sample belonging to the signal class versus the number of events from the background class. In the machine learning community techniques for learning from imbalanced data are well established. There is now even a software package, $\mathtt{imbalanced-learn}$ [8], dedicated to this task. In high energy physics there do not appear to be many cases where imbalanced learning techniques were explicitly used. However the measurement of the time-integrated $CP$ asymmetry in $D^0 \to K_S^0 K_S^0$ decays by LHCb [9] is one such example. In particular LHCb classified the $D^0$ decay signal from its background using the analysis methods developed in Refs. [10, 11]. An alternative approach to classification with imbalance techniques is using an anomaly detection framework. There are several examples of this in high energy physics [12–15].

Given the lack of examples where imbalanced learning techniques were used in high energy physics, the purpose of this note is two-fold. Firstly, in Section 2, we aim to provide a brief overview of modern class imbalance techniques in a high energy physics setting, introducing novel loss functions and a data resampling technique. Secondly, we provide two case studies of how class imbalance techniques can be used in high energy physics settings. The first case, presented in Sec. 3, is the measurement of the longitudinal polarization fraction in same-sign $WW$ scattering. We find a modest improvement in the performance of both the classical machine learning models and the deep learning models used in the longitudinal $WW$ study. The second study is the decay of the Higgs boson to charm-quark pairs, which follows in Sec. 4. Our Higgs-to-charm tagger gives a 14% improvement in the background rejection rate. Another application of these techniques is training directly on experimental data [16–18]. Conclusions are then given in Sec. 5. Much of the code for this project is available at [19].

# 2 Class Imbalance Techniques

There is no definitive answer to the question: What should one do when dealing with imbalanced data? The answer will depend on the data in question, see [20] for a study of benchmark datasets. In this Section we present a few approaches one might try to improve performance on an unbalanced dataset.

Using the accuracy of a classifier as a metric can be misleading. (See Table. 3 for a glossary of model evaluation terms used in this work.) Consider a model that predicts that every sample to be background. The accuracy of this model is $A = 1 - r$, where $r$ is the ratio of the number of signal events to the total number of events. Although this model would be highly accurate if the data were sufficiently imbalanced, it would not be useful as it says nothing about the signal, which is what

we were interested in to begin with. For this reason accuracy is not a recommended metric in this setting. The ROC curve is a good general purpose metric, providing information about the true and false positive rates across a range of thresholds, and the area under the ROC curve ($AUC$) is a good general purpose, single number metric. However, when dealing with imbalanced data, we argue in what follows that the precision-recall curve is the preferred metric to use on imbalanced data. If one instead prefers a single number metric, average precision is approximately the area under the precision-recall curve in analogy with $AUC$ for the ROC curve.

The ROC curve describes the false positive (background rejection) rate as a function of the true positive rate (signal efficiency), whereas the precision-recall curve, true to its name, gives precision as a function of recall. Recall is equivalent to the true positive rate, but precision does not correspond to the false positive rate. Recall or the true positive rate is a measure of how many true signal events have actually been identified as signal. Similarly the false positive rate is a measure of how many of the true background events have been identified as background. Precision, on the other hand, quantifies how likely an event is to truly be signal when a classifier has predicted it to be signal. A classifier's prediction will vary as the baseline probability of the positive class varies. As such, precision depends on how rare the signal is. This motivates using the precision-recall curve when the positive class samples are rare compared to the negative class examples. When this is not an issue the ROC curve is the metric to use as it does not care about the baseline probability of the positive class.

One might also try to balance the training set either by under-sampling [21–25] the majority class, oversampling the minority class [26, 27], or a combination of over- and under-sampling [28, 29]. Oversampling runs the risk of overfitting, and training with oversampling takes longer because of the additional data. For these reasons we will focus on under-sampling in this work. In particular, we will use random under-sampling to create a balanced random forest [30, 31]. Analogous procedures exist for creating a balanced boosted decision trees [32] and making balanced batches to feed into a neural network. The algorithm for how the balanced random forest makes classifications is as follows: (1) take bootstrap samples from the original dataset, (2) balance each sample by downsampling randomly, (3) learn a decision tree from each sample, (4) make predictions based on a majority vote. It is the second step of this process that is absent in a standard random forest. Even if this does not lead to a gain in performance training is faster with this approach because less data is used.

Lastly, one might consider making changes to the algorithms being used [33–35]. A simple example of this is if a metric such as precision, recall, or $F_1$ score is being used, its decision threshold can be optimized to maximize performance. One approach along these lines is to add hyperparameters to the loss function, creating a relatively larger penalty for misclassifying an example. To start consider the standard cross entropy loss function used for binary classification

$$BCE = -y \log(p) - (1-y) \log(1-p), \tag{1}$$

where $y$ is the ground-truth class with $y = 1$ for the signal class, and $p$ is the model's estimated probability that a given event belong to the signal class. Following Ref. [36] we introduce the following compact notation[1]

$$p_t = \begin{cases} p & \text{if } y = 1 \\ 1-p & \text{otherwise} \end{cases}. \tag{2}$$

With this definition Eq. (1) becomes

$$BCE = -\log(p_t). \tag{3}$$

---

[1] A model's estimated probability of an event belong to a class, $p_t$, is not to be confused with the transverse momentum of a particle, $p_T$.

When there is class imbalance it is common to add a weighting hyperparameter, $\alpha$, to loss function. Weighting the loss function can be implemented as follows

$$CE = -\alpha\, y \log(p) - (1-\alpha)(1-y)\log(1-p) \equiv -\alpha_t \log(p_t), \tag{4}$$

where $\alpha_t$ is defined analogously to $p_t$ in Eq. (2). With this normalization $\alpha$ takes values between 0 and 1. Often $\alpha$ is taken to be proportional to the inverse class frequency, $\alpha \propto r^{-1}$. The weighting hyperparameter balances the importance of signal and background events in the loss function. However $\alpha$ does not do anything to differentiate between easy- and hard-to-classify examples. In particular, easy-to-classify background examples may come to overwhelm the loss function even though they are individually negligible if the class imbalance is extreme enough.

This issue was rectified in Ref. [36], which introduced the focal loss function

$$FL = -(1-p_t)^\gamma \log(p_t), \tag{5}$$

where the modulating parameter, $\gamma$, puts the focus on hard-to-classify examples. In particular, when a sample is misclassified and $p_t$ is small the modulating factor is approximately one, and the loss is unaffected. However, as $p_t$ approaches one the modulating factor approaches zero, down-weighting the loss function for well-classified examples. When $\gamma = 0$ focal loss is equivalent to cross entropy, and as $\gamma$ is increased the rate at which easy-to-classify samples are down weighted also increases. Focal Loss is an optimal classifier just as cross entropy or mean square error are. One way to see this is Focal Loss produces a concave ROC curve (given sufficiently large statistics), which is equivalent to being optimized by the likelihood ratio [37].

In this work we will use the weighted variation of focal loss

$$FL = -\alpha_t(1-p_t)^\gamma \log(p_t), \tag{6}$$

with default values for the hyperparameters, $\alpha = 0.25$ and $\gamma = 2$.

Another generalization of focal loss is from binary classification to multi-class classification. Here the compact $p_t$ notation does not work, so to set the stage we define the categorical cross entropy loss for classification with $K$ classes

$$CCE = -\sum_{i=1}^{K} y_i \log(p_i), \tag{7}$$

where $p_i$ the probability that an example belongs to class $i$, and is given by the softmax function

$$p_i = \frac{e^{s_i}}{\sum_{j=1}^{K} e^{s_j}}, \tag{8}$$

with $s_i$ being the score for the $i$th class for an example. The vector $y$ is a one-hot representation of the classes with one component equal to one and the remain $K-1$ components equal to zero. When $K = 2$ Eq. (8) reduces to Eq. (3). With all of the setup in place, we can now write the categorical focal loss for multi-class classification

$$CFL = -\sum_{i=1}^{K} y_i(1-p_i)^\gamma \log(p_i). \tag{9}$$

# 3 Longitudinal Polarization Fraction in Same-Sign $WW$ Production

## 3.1 Introduction

Same-sign $WW$ production at the LHC is the vector boson scattering (VBS) process with the largest ratio of electroweak-to-QCD production. As such it provides a great opportunity to study whether the discovered Higgs boson leads to unitary longitudinal VBS, and to search for physics beyond the SM (BSM) [38, 39]. The ATLAS and CMS experiments have observed electroweak same-sign $WW$ production in the two jet, two same-sign lepton final state in 13 TeV $pp$ collisions with significances of $6.9\sigma$ [40] and $5.5\sigma$ [41], respectively. Confirming or refuting the unitarity of VBS requires not just a measurement of $pp \rightarrow jjW^{\pm}W^{\pm}$, but of the fraction of these events where both $W$s are longitudinally polarized ($LL$ fraction).

Prospects for the extraction of the longitudinal component of $W^{\pm}W^{\pm}$ scattering during the High-Luminosity phase of the LHC (HL-LHC) were studied in Refs. [42–44]. The fraction of longitudinally polarized events is predicted to be only $r \sim 0.07$ in the SM at large dijet invariant mass ($m_{jj}$) [43] making this a challenging measurement. Using the difference in the azimuthal angle of the two jets

$$\Delta\phi_{jj} = \min(|\phi_{j_1} - \phi_{j_2}|, 2\pi - |\phi_{j_1} - \phi_{j_2}|), \tag{10}$$

as a discriminant, the significance for the observation of the $LL$ fraction is expected to be up to $2.7\sigma$ with 3000 fb$^{-1}$ of integrated luminosity [43].

The observation significance can be improved through the use of deep learning [45, 46]. Ref. [45] regressed on the angles between the charged leptons in their parent boson's rest frame and the $W$ boson's direction of motion, whereas Ref. [46] treated this as a binary classification problem distinguishing between events where both $W$s were longitudinally polarized versus when one or none of the $W$s were polarized. In the classification setting it is important to keep in mind that the predicted $LL$ fraction is small, and thus there is an imbalance in the number of events belonging to the class $N(W_L) = 2$ versus the class $N(W_L) < 2$ ($LL$ class vs. $TL + TT$ class). We proceed treating this as a classification problem with imbalanced classes.

## 3.2 Data

`MadGraph5` v2.6.6 [47] is used to simulate events for the leading order electroweak, $\mathcal{O}(\alpha^4)$, contribution to process $pp \rightarrow jjW^{\pm}W^{\pm}$ at center of mass energy $\sqrt{s} = 14$ TeV. The fraction of events where both $W$s are longitudinally polarization is $r \approx 7.5\%$. Additionally, `MadSpin` [48] is used to include spin correlation effects in the decays of the $W$ bosons such that the final process under consideration is $pp \rightarrow jj\ell^{\pm}\nu\ell^{\pm}\nu$ with $\ell = \{e, \mu\}$. Representative Feynman diagrams are given in Figure 1. Note that in this case study, unlike the one that follows it, the "jets" are partons from the hard scattering process and are not showered or hadronized. We comment on the impact this choice has in the results subsection of this case study. The cuts are chosen to match those of Ref. [46]. We require two jets with transverse momentum, $p_T > 50$ GeV, and pseudorapidity, $|\eta| < 4.7$. The jet pair must also have an absolute difference in pseudorapidity $\Delta\eta_{jj} > 2.5$, consistent with VBS, and have an invariant mass $m_{jj} > 850$ GeV to suppress non-prompt and $WZ$ backgrounds [41]. Additionally we select for two same-sign charged leptons with $p_T > 20$ GeV and $|\eta| < 2.4$. A total of approximately $1.7 \cdot 10^5$ events pass these cuts.

The feature engineering is also done to match that of Ref. [46] as much as possible. The $p_T$, $\eta$, and $\phi$ of the two jets and the two leptons are used as features. The subscripts 1 and 2 are used to indicate the jet or lepton with the larger or smaller transverse momentum, *e.g.* $p_T^{j_1} > p_T^{j_2}$. This step

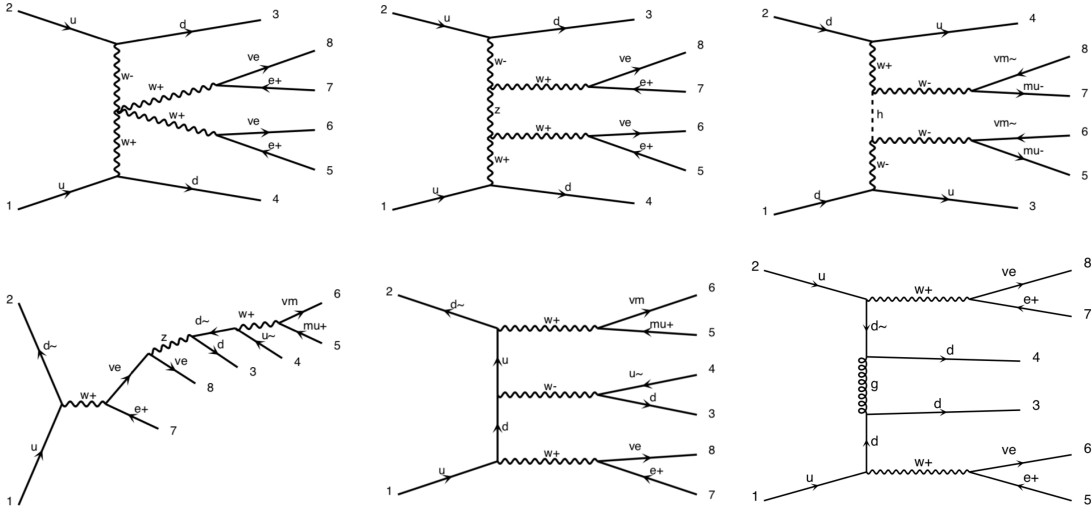

Figure 1: Representative leading order Feynman diagrams for $pp \to jj\ell^{\pm}\nu\ell^{\pm}\nu$. Top row: Diagrams contributing to the signal, $pp \to jjW^{\pm}W^{\pm} \to jj\ell^{\pm}\nu\ell^{\pm}\nu$ with $\sigma \propto \alpha^6$. Bottom row: Diagrams considered irreducible background in this work. The diagrams are drawn with MadGraph5 [47].

that improves the performance of classifiers, and is not done by default in MadGraph5. The magnitude and azimuthal angle of the missing transverse energy are included as well. In addition, the following high-level features are added. From the jet system we add the invariant mass, the difference in pseudorapidity, and the difference in the azimuthal angle. We also consider the Zeppenfeld variable [49] for the two charged leptons,

$$z_{\ell_i} = \frac{\eta_{\ell_i} - \bar{\eta}_{jj}}{\Delta\eta_{jj}}, \tag{11}$$

where $\bar{\eta}_{jj}$ is the mean pseudorapidity of the two leading jets. Finally we include the separation of the di-jet and di-lepton systems in the pseudorapidity-azimuthal angle plane, $\Delta R_{jj,\ell\ell}$, bringing the total number of features to 20.

## 3.3 Models and Training

In addition to using $\Delta\phi_{jj}$ and $p_T^{\ell_1}$ as discriminating observables, we use the following models. For classical machine learning we use a random forest (RF) as a baseline, and look to use a change in performance from weighting or balancing. We use the $\mathtt{imbalanced-learn}$ [8] implementation of balanced random forest, and use $\mathtt{scikit-learn}$ [50] for the other random forests. The balanced random forest has no maximum depth, while the other random forests have a maximum depth of 10. Additionally we consider a LightGBM [51] (LGBM), which is a gradient boosted decision tree where the trees are grown in a depth first rather than breadth first fashion. The name Light comes from the fact that the training time is often greatly reduced with this construction of the trees. In particular, our LGBM has $10^3$ estimators and a learning rate of 0.01. The deep learning models are fully-connected neural networks (DNNs) implemented using the Keras API [52] for TensorFlow v2.0.0 [53]. Our baseline DNN has a cross entropy loss function, Eq. (3), and the variation we test is

Table 1: Results of the five-fold cross validation for classifying $LL$ events from $TL + TT$ events in $pp \rightarrow jjW^{\pm}W^{\pm} \rightarrow jj\ell^{\pm}\nu\ell^{\pm}\nu$. Performance is reported as (the mean $\pm$ the standard deviation) of the five folds. $t_{\text{fit}}$ is the time it takes to fit the model to a training fold of data. The models utilizing class imbalance techniques show modest improvements in performance with respect to their baselines. See the text more for details.

| Model | $t_{\text{fit}}$ [s] | Average Precision | $AUC$ |
|---|---|---|---|
| $\Delta\phi_{jj}$ | - | $0.120 \pm 0.003$ | $0.662 \pm 0.006$ |
| $p_T^{\ell_1}$ | - | $0.112 \pm 0.003$ | $0.663 \pm 0.006$ |
| Random Forest | $84 \pm 24$ | $0.223 \pm 0.006$ | $0.766 \pm 0.006$ |
| Weighted RF | $30 \pm 15$ | $0.227 \pm 0.006$ | $0.768 \pm 0.006$ |
| Balanced RF | $63 \pm 19$ | $0.228 \pm 0.007$ | $0.776 \pm 0.005$ |
| LightGBM | $9.7 \pm 0.7$ | $0.241 \pm 0.005$ | $0.782 \pm 0.005$ |
| Deep Neural Network | $(2.8 \pm 0.3) \cdot 10^2$ | $0.244 \pm 0.008$ | $0.789 \pm 0.004$ |
| DNN w/ Focal Loss | $(3.3 \pm 1.1) \cdot 10^2$ | $0.246 \pm 0.004$ | $0.791 \pm 0.005$ |

a DNN with a focal loss function, Eq. (6). The features are scaled to have zero mean and unit variance before being fed into the neural networks. All of our neural networks have 2 hidden layers each with 150 neurons, He initialization, and ReLU activation functions. Batch normalization is performed to speed up the learning process, dropout is applied at a 50% rate for regularization, and the Adam algorithm is used to optimize the parameters of the DNN.

A five-fold cross validation is performed for each for model. The folds are stratified based on the size of the class imbalance. For the DNNs, a batch size of 50 is used in training. Early stopping is implemented for the DNNs where training runs until there is no decrease in the training loss function for 5 consecutive epochs. Similarly, we grow the Random Forests 10 trees at a time until there is no improvement in the training loss function.

## 3.4 Results

Table 1 shows the results of the cross validation with performance being reported as the mean $\pm$ the standard deviation of the five folds. Both the weighted random forest and the balanced random forest modestly outperform the baseline random forest. Similarly, the DNN with focal loss modestly outperforms its baseline neural network. The uncertainty on the machine learning metrics is statistical in nature; one over the square root of the sample size of a test fold in the cross validation is approximately $5.4 \cdot 10^{-3}$. On the other hand, the uncertainty on the time it takes to fit the models, $t_{\text{fit}}$, does not follow this statistical pattern due to the stochastic nature of the optimization process and the early stopping criteria imposed on training.

The improvement in performance of the balanced RF can be seen visually in Figure 2 where the green curves of the standard random forest are below the red curves of the balanced random forest both precision versus recall (left panel) and the ROC curve (right panel). More strikingly, all of the machine learning models significantly outperform the kinematic variable $p_T^{\ell_1}$. Note that recall is equivalent to signal efficiency, but precision is not related to background rejection.

The balanced and weighted random forests also take less time to train. In the case of the balanced RF, $t_{\text{fit}}$ does not tell the whole story as it has no maximum depth whereas the standard random forest can only be 10 levels deep. Not to be outdone, the LGBM fits more than an order of magnitude faster

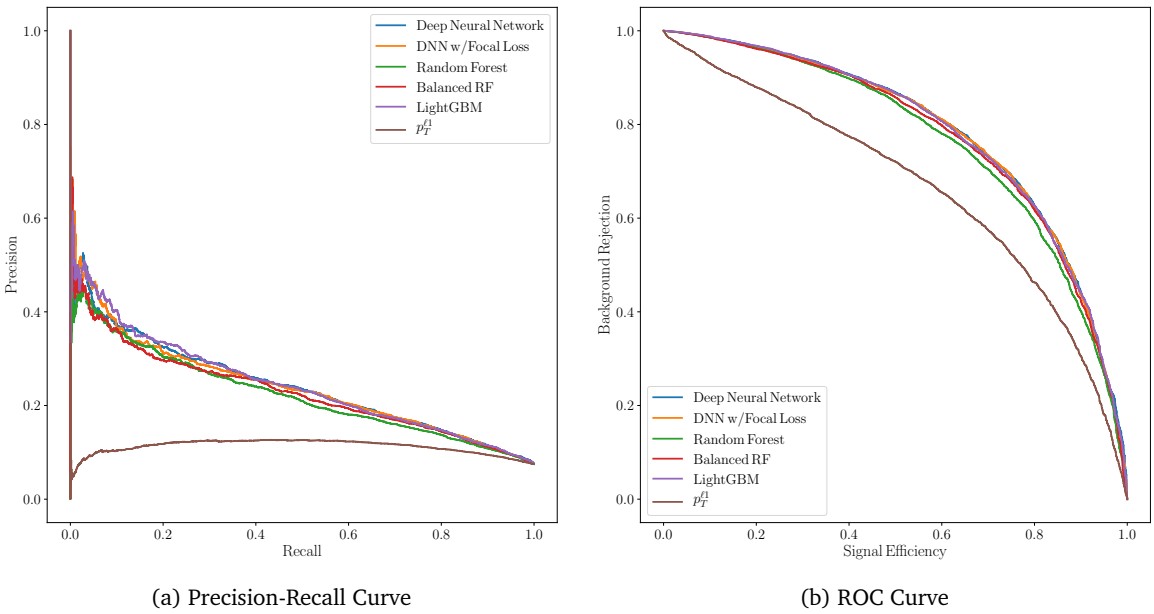

(a) Precision-Recall Curve      (b) ROC Curve

Figure 2: Multiple parameter performance measures. The precision-recall curve is given in 2a, and the ROC curve is shown in 2b. Visually it is clear that the balanced random forest (red) outperforms its unbalanced counterpart (green). More strikingly, all of the machine learning models significantly outperform the kinematic variable $p_T^{\ell_1}$. Note that recall is equivalent to signal efficiency, but precision is not related to background rejection.

than the neural networks and almost an order of magnitude faster than the standard random forest. Its performance is intermediate between the balanced random forest and the baseline the neural network.

Histograms for the probability the event will be predicted to be an $LL$ event are shown in Figure 3 when it is in truth an $LL$ event (red distributions) or when it is actually an $TL + TT$ event (blue distributions). The top row shows the random forest models, and the bottom row shows the DNN models.

The mean predicted probability for a classifier with an unweighted loss function trained on an imbalanced dataset is $r$, the imbalance ratio. Complete signal-background separation in the training dataset is a sign of overfitting if such behavior is not also observed in the validation dataset, which it's not in this case. Balancing the training set moves the mean value from $r$ to 0.5. This can be seen in the upper right panel of 3 from the balanced random forest. Weighting the loss function with the inverse of the class frequencies also moves the mean value to 0.5. Focal loss is intermediate between these two scenarios, $r$ and 0.5, as can be seen in the bottom right panel of 3.

Finally, this case study would not be complete without a comparison with to Ref. [46]. The most obvious difference between our work and that of [46] is the better performance we find from the kinematic variable $\Delta\phi_{jj}$. However we did not pass our simulated events through a parton shower or hadronize them, which likely would have spoiled some of the correlation between $\Delta\phi_{jj}$ and the polarizations of the $W$ bosons. Beyond that, our results are consistent with those found in Ref. [46]. Specifically, as measured by the $AUC$, our fully-connected neural network with two hidden layers

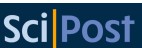

Figure 3: Histograms for the probability the event will be predicted to be an $LL$ event when it is in truth an $LL$ event (red distributions) or when it is actually an $TL + TT$ event (blue distributions). The top row shows the random forest models, and the bottom row shows the DNN models.

matches the performance of the neural network with "particle-based" architecture and 10 hidden layers in [46]. Additionally, our balanced random forest matches the performance of the AdaBoost classifier of Ref. [46], where again performance is measured by the $AUC$. We do not estimate the statistical significance of a non-zero $LL$ fraction from our classifiers for two reasons. Firstly the imbalance ratio $r$ is higher in our simulated dataset than that of Ref. [46], which would make our models appear to significantly outperform those of [46] when based on the comparison of machine

learning metrics given above the differences are not so great. Secondly all the machine learning models significantly outperform the kinematic variable $p_T^{\ell_1}$, as can be seen in Fig. 2, so it's safe to assume all of the models tested here would produce a significance similar to $5\sigma$ given that the neural network in [46] was able to do so.

# 4 Higgs Boson Decays to Charm-Quark Pairs

## 4.1 Introduction

The second application of class imbalance techniques we explore in this note is to the measurement of Higgs boson decays to charm-quark pairs. Searches for the decay of the Higgs boson to charm-quarks have produced only weak limits to date. ATLAS reported an upper limit of 110 times the SM rate for the process $pp \to Zh \to \ell^-\ell^+c\bar{c}$ [54]. LHCb instead considered the associated production of both $W$s and $Z$s in range $2 < \eta < 5$, and set a limit of 6,400 times the SM rate [55]. A result of these weak limits is that direct limits on the charm Yukawa coupling are correspondingly weak. Stronger bounds can be obtained indirectly, *e.g.* through global fits [56–64], among other methods.[2] However there are assumptions build into any indirect analysis. The limit on the charm Yukawa coupling at HL-LHC is projected to get down to about 2.2 times the SM rate [63] (see also [66]). Based on this projection an observation of $h \to c\bar{c}$ is not expected at HL-LHC motivating ways to improve the analysis, although this projected limit should still be useful in constraining certain BSM physics.

One reason for the weak limits on $h \to c\bar{c}$ is in the SM the rate for $h \to b\bar{b}$ is about 20 times larger ($r \approx 0.05$) than the rate for $h \to c\bar{c}$ [67]. In contrast with $h \to c\bar{c}$, the decay of the Higgs boson to bottom-quarks has been observed by both ATLAS [68] and CMS [69] The analyses of Refs. [54, 55, 68, 69] rely on tagging the flavor of the jets, which involves discriminating charm initiated jets from bottom jets, or vice versa, and discriminating heavy from light flavored jets.[3] The use of flavor tagging explicitly links the measurements of $h \to b\bar{b}$ and $h \to c\bar{c}$ [71, 72].

To perform the flavor tagging LHCb used their standard, state-of-the-art heavy flavor tagger [73], while ATLAS trained boosted decision trees to separate charm from light jets and charm from bottom jets with a procedure analogous to how they train their standard bottom tagger [74, 75]. The use of general purpose flavor tagging algorithms is less then ideal for the specific task of identifying Higgs decays to charms. This was recognized in Ref. [76], which made a dedicated double-charm tagger for $h \to c\bar{c}$. We also advocate making a dedicated $h \to c\bar{c}$ tagger for the following reason. The standard heavy flavor tagging algorithms are not optimized for the imbalance in the expected number of $h \to c\bar{c}$ versus $h \to b\bar{b}$ events. For example, QCD produces roughly equal numbers of bottoms and charms at invariant masses relevant for Higgs physics. Given the statistical nature of heavy flavor tagging, an imbalance in the number of $b\bar{b}$ and $c\bar{c}$ decays will lead to worse performance in identifying the Higgs to charm events. As such this is a well motivated arena for applying class imbalance techniques. Here we are assuming a SM-like rate for $h \to c\bar{c}$. If some BSM physics makes the experimental rate for $h \to c\bar{c}$ much larger than expected this would invalidate our argument (which would be a small price to pay for the discovery of the breakdown of the SM). The rest of this case study delivers proof of principle that it is possible to improve tagging efficiency of $h \to c\bar{c}$ events through the use of the class imbalance techniques.

Looking beyond the proof of principle, a few additional steps to be taken in future work are described in what follows. We are treating this as a binary classification problem of distinguishing

---

[2]The special role in global fits of the Higgs boson coupling to charm-quarks has been known for a long time [65].

[3]A complementary approach is to exclusively search for charmed-hadrons [70].

Higgs boson decays to charm-quark pairs from bottom-quark pairs. Firstly, extending our approach to also discriminate heavy flavor jets from light flavor jets will make our tagger more like what the experiments are currently doing. A second opportunity area stems from our study of charm-tagging at a lepton collider where experimental tagging might not be based on jets, while it's clear that at hadron colliders jet based analyses are and will continue to be used. Lastly, a direct comparison with the results Ref. [76] is not currently possible given the different background considered in the two works. It would be useful to do a proper comparison of the two tagging methods.

## 4.2 Data

We consider associated Higgs production at an $e^+e^-$ collider as an observation of $h \to c\bar{c}$ is not expected at HL-LHC. Specifically, the process under consideration is $e^+e^- \to Zh \to \ell^+\ell^- Q\bar{Q}$ with $\ell = e$ and $\mu$, and $Q = b$ or $c$. A total of $2 \cdot 10^5$ events are simulated with MadGraph5 [47] with Pythia6 [77] used for parton showering and hadronization. Half the simulated events are $h \to b\bar{b}$ and the other half are $h \to c\bar{c}$. We focus on the binary classification problem of $h \to c\bar{c}$ versus $h \to b\bar{b}$ as existing tagging algorithms perform well at distinguishing heavy from light flavors, see *e.g.* [73]. The center-of-mass energy of the collisions is $\sqrt{s} = 250$ GeV. Jets are clustered using the FastJet [78] implementation of the anti-$k_t$ clustering algorithm [79] with radius parameter $R = 0.4$. We require at least two jets each with $p_T > 10$ GeV. Similarly, we require the leptons to be oppositely charged, and to each have $p_T > 10$ GeV.

The four-vector of each lepton and the two leading jets are used as features. In particular we use the mass, $m$, of the jet or lepton as a feature. It is unlikely that the mass of a jet could be measured with enough precision in an actual experiment to distinguish a charm initiated jet from a bottom jet. However the mass of the jet is a proxy for the lifetime of the initiating particle of the jet, which is a feature flavor tagging algorithms exploit, see *e.g.* [54]. The four-vectors of the dilepton and dijet systems, which reconstruct the $Z$ and Higgs bosons, respectively, are also included in our feature set. A cut on the invariant mass of the jets is imposed, $95 < m_{jj}/\mathrm{GeV} < 155$, to concentrate on resonant Higgs production. All of the above cuts and requirements reduce the number of simulated events to approximately $8.9 \cdot 10^4$. We include

$$\Delta R = \sqrt{(\eta_{j1} - \eta_{j2})^2 + (\phi_{j1} - \phi_{j2})^2} \tag{12}$$

between the two jets as a feature as well as the rescaled mass drop observable, $ISY$, and the radius of the dijet system, $R_{jj}$,

$$ISY = \frac{\max(m_{j1}, m_{j2})\Delta R}{m_{jj}}, \quad R_{jj} = \frac{m_{jj}(p_{T,j1} + p_{T,j2})}{p_{T,jj}\sqrt{p_{T,j1}p_{T,j2}}}. \tag{13}$$

Lastly, as bottom- and charm-quarks are oppositely charged, we look at the charge of the jets as defined in [80]

$$\mathcal{Q}_\kappa^j = \frac{1}{(p_{T,j})^\kappa} \sum_{p \in j} Q_p (p_{T,p})^\kappa, \tag{14}$$

where the charge, $\mathcal{Q}$, of a jet, $j$ is the $p_T$ weighted sum of the charges, $Q$, of all the partons, $p$, in the jet. We use $\kappa = 0.4$ in this work. Of course only the overall magnitude of the jet charges differ between bottom and charm Higgs decays. Therefore, in addition to the charge of each jet, we include the product of the jet charges, the absolute value of the difference of the jet charges, and the charge of the dijet system, bringing our total number of features to 30.

## 4.3 Models and Training

Our heavy flavor tagging model is a `LightGBM` [51]. In particular, our model combines a mere 50 trees in series, and each tree is allowed to have a maximum depth of 10 with all other hyperparameters fixed to their default values. We take as our baseline heavy flavor tagger a `LightGBM` with an unweighted loss function, and compare its performance against a `LightGBM` with weighting $\alpha = 1 - r$.

For model evaluation we again perform a stratified five-fold cross validation. We test three scenarios. In the first test we assume the rate for $h \to c\bar{c}$ is equivalent to the rate for $h \to b\bar{b}$. Here we use the baseline LGBM with unweighted loss function. In this case there is no class imbalance implying there must be some BSM physics in this scenario. We randomly select $4.0 \cdot 10^4$ bottom and $4.0 \cdot 10^4$ charm events from our full simulated dataset, and perform the cross validation on this sample. For the second test we again use the unweighted, baseline model, but perform the cross validation on dataset with SM-like class imbalance. In particular we randomly select $4.0 \cdot 10^4$ bottom and $2.0 \cdot 10^3$ charm events from our full simulated dataset. For the third and final test we reuse the dataset from the second test, but use our class imbalance optimized LGBM with weighting hyperparameter $\alpha = 1 - r \approx 0.95$.

## 4.4 Results

The results of our three $h \to c\bar{c}$ tagging tests are given in Table 2 with the rows from top to bottom corresponding to the 1st, 2nd, and 3rd scenarios described in previous subsection. For each scenario we consider two signal efficiency working points, a looser selection of $\epsilon_{h \to c\bar{c}} = TPR = 0.2$ and a tighter selection of $\epsilon_{h \to c\bar{c}} = 0.8$. We report the background rejection rate, $\epsilon_{h \to b\bar{b}} = FPR$, for each of these working points. The inverse of the background rejection rate is largest in the scenario without class imbalance. The performance of both tagging models is worse in the presence of class imbalance. However the weighted LGBM outperforms the baseline tagging model in the presence of class imbalance, demonstrating proof of principle that class imbalance techniques can be used to improve the performance of algorithms used to identify $h \to c\bar{c}$ events. In particular, there is a 14% increase in $1/\epsilon_{h \to b\bar{b}}$ with loose selection criteria when the class imbalance optimized model is used.

We also report the average precision, and average precision normalized by the imbalance ratio. The average precision is significantly higher in the scenario without class imbalance. However when the average precision is normalized by the imbalance ratio, which constitutes the naïve expectation for the *AP* score, higher values are found when the data is imbalanced.

Additionally, Fig. 4 shows the precision-recall curves for our three $h \to c\bar{c}$ tagging tests. The blue, orange, and green curves correspond to the test results in the top, middle, and bottom rows of Table 2, respectively. These curves provide another way of demonstrating that the weighted LGBM outperforms (green) outperforms its unweighted counterpart (orange). Specifically, at lower recall, weighting the loss function to remove class imbalance leads to a gain in performance. Recall is equivalent to true position rate or signal efficiency, $\epsilon_{h \to c\bar{c}}$.

Lastly, we investigate which features are important for the classification. Using the feature importance of the LGBM the charges of the heavy flavor jets and the associated engineered features do not play a significant role in discriminating charm initiated jets from bottom jets. This is in contrast with studies of light flavored jets [81]. A possible explanation for this is the heavy flavored hadrons have more possible decay chains.[4] In particular, a neutral meson may oscillate or there might be a cascade decay that spoils the correlation between the charges of the partons in the jet and the charge

---

[4]This is one of the main systematic uncertainties in measuring asymmetric heavy quark hadroproduction [82–85].

Table 2: The results of our three $h \to c\bar{c}$ tagging tests. We report the background rejection rate, $\epsilon_{h \to b\bar{b}} = FPR$, for two signal efficiency working points, $\epsilon_{h \to c\bar{c}} = TPR = 0.2(\text{loose}), 0.8(\text{tight})$. There is a 14% increase in $1/\epsilon_{h \to b\bar{b}}$ with loose selection criteria when the class imbalance optimized model is used, 3rd versus 2nd row, demonstrating proof of principle that class imbalance techniques can be used to improve the performance of algorithms used to identify $h \to c\bar{c}$ events. We also report the $AP$, and $AP/r$, with the latter given to one decimal place for better readability.

| Model | $\alpha$ | $r$ | $\epsilon_{h \to c\bar{c}}$ | $1/\epsilon_{h \to b\bar{b}}$ | Average Precision | $AP/r$ |
|---|---|---|---|---|---|---|
| LightGBM | $\frac{1}{2}$ | 1 | 0.2 | $38.8 \pm 2.6$ | $0.719 \pm 0.004$ | $0.7 \pm 0.0$ |
| | | | 0.8 | $1.7 \pm 0.0$ | | |
| LightGBM | $\frac{1}{2}$ | 0.05 | 0.2 | $30.9 \pm 5.2$ | $0.166 \pm 0.008$ | $3.3 \pm 0.2$ |
| | | | 0.8 | $1.6 \pm 0.1$ | | |
| Weighted LGBM | $1-r$ | 0.05 | 0.2 | $35.1 \pm 8.9$ | $0.161 \pm 0.011$ | $3.2 \pm 0.2$ |
| | | | 0.8 | $1.5 \pm 0.1$ | | |

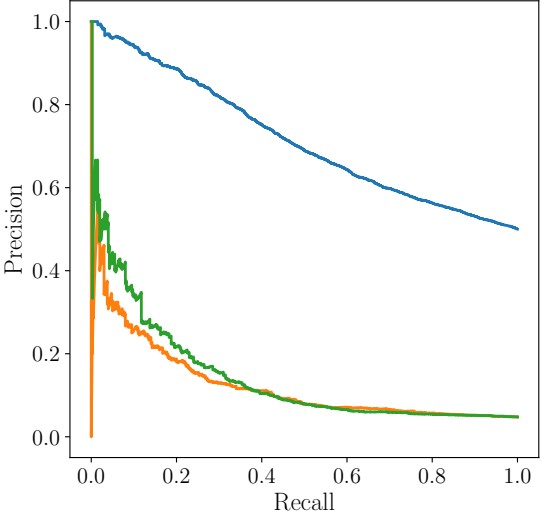

Figure 4: The precision-recall curves for our three $h \to c\bar{c}$ tagging tests. The blue, orange, and green curves correspond to the test results in the top, middle, and bottom rows of Table 2, respectively. These curves provide another way of demonstrating that the weighted LGBM outperforms (green) outperforms it unweighted counterpart (orange). Specifically, at lower recall, weight the loss function to remove class imbalance leads to a gain in performance. Recall is equivalent to true position rate or signal efficiency, $\epsilon_{h \to c\bar{c}}$.

of the particle that initiated the jet. Again using the feature importance of the LGBM, we find the the four-vectors of the leptons and the four-vector of the reconstructed $Z$ boson also do not play a major role in discriminating charm initiated jets from bottom jets.

# 5 Discussion

Extracting a signal from a much larger background is a common problem in high energy physics. Posed as a classification task, there is said to be an imbalance in the number of samples belonging to the signal class versus the number of samples from the background class. Imbalanced learning techniques are not commonly used, explicitly anyways, in high energy physics. Given this lack of use we first provided a brief overview of modern class imbalance techniques in a high energy physics setting, introducing novel loss functions and a data resampling technique. We then presented two case studies illustrating these techniques. The first study is the measurement of the longitudinal polarization fraction in same-sign $WW$ scattering. We found a modest improvement in the performance of both the classic ML models and in the deep learning models tested in the longitudinal $WW$ study. Our neural networks achieves comparable performance to that of Ref. [46] despite having only two hidden layers instead of 10. Given that there are only $\mathcal{O}(10)$ features in this dataset it is not surprising that a very deep network did not continue to improve performance. Having fewer hidden layers with all else being equal results in a reduction in training time. The second case is the decay of the Higgs boson to charm-quark pairs. We delivered proof of principle that it is possible to improve tagging efficiency of $h \to c\bar{c}$ events through the use of the class imbalance techniques. In particular, our Higgs-to-charm tagger with loose selection criteria gave a 14% improvement in the background rejection rate.

## Acknowledgements

We thank Eder Izaguirre and Brian Shuve for collaborations on a previous incarnation of the $h \to c\bar{c}$ tagging study. We would also like to thank Emmanuel Ameisen, Sally Dawson, Samuel Homiller, and Marc-André Pleier for useful discussions. Finally, we are grateful to Tilman Plehn and the anonymous referee for many helpful comments on the manuscript.

# A  Glossary

See Table 3 for a glossary of model evaluation terms used in this work.

Table 3: Glossary of model evaluation terms used in this work.

| Metric | Symbol | Definition |
|---|---|---|
| Accuracy | $A$ | $A = (TP + TN)/(FN + FP + TN + TP)$ |
| Area Under the ROC Curve | $AUC$ | $AUC = \int_0^1 d(TPR)[1 - FPR(TPR)]$ |
| Average Precision | $AP$ | $AP = \sum_n (R_n - R_{n-1})P_n$ |
| Decision Threshold | $n$ | if $p > n$ for a given event, then that event is predicted to be signal |
| F1 score | $F_1$ | $F_1 = 2P \cdot R/(P + R)$ |
| False Negative | $FN$ | a signal event that is predicted to be background |
| False Positive | $FP$ | a background event that is predicted to be signal |
| False Positive Rate | $FPR$ | $FPR = FP/(FP + TN)$ |
| Ground Truth Class | $y$ | $y = 1$ if the event is truly a signal event, and $y = 0$ if it is background |
| Precision | $P$ | $P = TP/(FP + TP)$ |
| Probability Estimate | $p$ | a model's estimated probability that a given event belongs to the signal class |
| Recall | $R$ | $R = TP/(FN + TP)$ |
| True Negative | $TN$ | a background event that is predicted to be background |
| True Positive | $TP$ | a signal event that is predicted to be signal |
| True Positive Rate | $TPR$ | $TPR = R$ |

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
