# Peer review of "Class Imbalance Techniques for High Energy Physics"

_SciPost Physics, doi:SciPost Phys. 7, 076 (2019)_

## Round 1 · Referee Report · Tilman Plehn (Referee 1) · 2019-9-11

Strengths

1- Generally, the paper has something significant to add to the very modern field of machine learning; 2- Class imbalance is not yet widely used and the related opportunities should be discussed more in particle physics; 3- I love the idea that former particle theorists keep thinking about particle physics and publish papers based on their developing expertise.

Weaknesses

1- The author is starting to use ML slang at a level where general particle theorists might not be able to follow; 2- Some of the presentation and explanations should be improved (see comments below)

Report

Definitely worth publishing, because it contains a nice mix of relevant physics problems and technical sophistication. But the paper could be more clear and (even) more useful if it provided more details and gave more quantitative results. As it stands it does not actually encourage people like me to actually use these new ideas, and I would very much like to be convinced.

Requested changes

Ordered by appearance in the paper, not by relevance: 1- please elaborate on the precision-recall curve, which unlike ROC is not generally known in particle physics; 2- The discussion of the loss functions Eqs.(1-3) is a little brief for non-experts, for example for cases like H->cc I would also like to hear something about multi-class cases; 3- Footnote 1 is pure slang; 4- Sec.3 would benefit from a set of sample Feynman diagrams with the different signal and background processes; 5- I am a big fan of more than one-parameter measures in serious science, please add for instance an ROC curve or whatever works best to Tab.1; 6- The comparison to Ref.[46] needs some kind of plot or curve or number. Is the mjj cut related to controlling otherwise problematic backgrounds? 7- What is the problem with the signal and background separation in the DNN panels of Fig.1? Please discuss, it does not look good; 8- How are the error bars in Tab.1 obtained? Naively, the last column seems to indicate that the improvement is not really significant. Please discuss this in more detail, after all we are particle physicists with a famous fetish for error bars; 9- The reference to [53] looks too much like an ad-hoc self-citation. Please add the other global analyses, for instance of Run II data. The special role of the Hcc coupling in a global fit is long known, at least since 0904.3866 and its Sec.4.3. 10- How close is the charm tagging described in Sec.4.2 to what experiments do? It's not clear to me. Moreover, are we sure that an ILC analysis would be based on jets to begin with, and not on particle flow objects, tracks, etc? 11- As before, Tab.2 is interesting, but one parameter is too naive a measure for particle physics, please provide more information and more discussion. 12- Any chance you can estimate the performance of Ref.[65] compared to the new tagger approach?

  • validity: high
  • significance: high
  • originality: high
  • clarity: good
  • formatting: excellent
  • grammar: excellent

Author:  Christopher W. Murphy  on 2019-11-24  [id 654]

(in reply to Report 1 by Tilman Plehn on 2019-09-11)
Category:
answer to question
reply to objection
validation or rederivation

Hi Tilman,

Thanks for the encouraging report! I have made the following changes to my manuscript in response to the points you raised.

1.) I revised a paragraph in Section 2 discussing the precison-recall curve so that it goes into more detail.
2.) I added several paragraphs to Section 2 going into greater detail on focal loss and what it is useful for. In a nutshell it down weights easy-to-classify examples in the loss function so that training is more focused on the hard-to-classify examples.
3.) What was footnote 1 has been removed
4.) Feynman diagrams have been added, see Figure 1
5.) Figure 2 has been added, which gives both the precision-recall curve and the ROC curve for the models in Table 1.
6.) The cuts and feature engineering have been updated to match for Ref. [46] as closely as possible. Good agreement is found when a fair comparison can be made. I find the AUC to be higher for $\Delta\phi_{jj}$ than Ref. [46]. However this is due to my treatment of partons from the hard scattering process as jets.
7.) There is no problem with the signal-background separation in (what is now) Figure 3. The mean predicted probability for a classifier with an unweighted loss function trained on an imbalanced dataset is $r$, the imbalance ratio. I would take complete signal-background separation in the training dataset to be a sign of overfitting if such behavior was not also observed in the validation dataset, which it’s not in this case. Balancing the training set moves the mean value to 0.5. This can be seen in the upper right panel from the balanced random forest. Weighting the loss function with the inverse of the class frequencies also moves the mean value to 0.5. Focal loss is intermediate between these two scenarios, $r$ and 0.5, as can be seen in the bottom right panel.
8.) Performance is reported as the mean $\pm$ the standard deviation of the five folds of the cross validation. I added this to the caption of the table. It was already in the text. My apologies for the confusion around the use of the word significant. This is certainly not a Higgs boson level discovery, nor even 3-sigma evidence for something. Apparently I’ve been out of physics for too long. I was approaching from of the point of view how large of an effect would need to be observed to convince my business stakeholders it would be worth rolling this out. The manuscript has been revised accordingly.
9.) The global fit reference now includes citations for 9 papers, Ref. [56-64], and I added a footnote about 0904.3866.
10.) I added a paragraph to end of Section 4.1 attempting to clarify these questions.
11.) The precision-recall curves for the tests in Table 2 have been added in Figure 3. The curves are consistent with the finding in Table 2, and provide another way of demonstrating that class imbalance techniques can lead to a performance in charm tagging efficiency.
12.) A direct comparison against (what is now) Ref. [76] is not possible given the backgrounds considered in the two works are different.

Sincerely,
Chris

---

## Round 1 · Referee Report · Anonymous (Referee 2) · 2019-9-23

Report

The present manuscript explores the new Imbalance Technics in Machine Learning for the common classification problem in High Energy physics of signal extraction from a much larger background. The proposed study applies this technic to two well-motivated physics measurements: the longitudinal polarization fraction measurement in same-sign WW production and branching ratio measurement of the Higgs boson to charm-quark pairs. The study finds relevant improvements in performance with regard to the classic Machine Learning models. While the paper is certainly worth publishing, the present manuscript exhibits some points that would benefit from further improvements.

Requested changes

1) The longitudinal polarization study for the same-sign WW production does not account for the W decays. I recommend that the author includes the W decays and performs the Machine Learning analysis using the charged lepton observables instead of the W-boson momentum. The more realistic observables will further improve this study making the quoted significances and the comparison to the simple $\Delta \phi_{jj}$ in Tab. 1 and Ref. [46] more reliable.

2) To make the final results more robust, I would suggest performing the same-sign WW analysis with a more restrictive threshold on $m_{jj}$. The adopted threshold is too low and significantly enhances the non-prompt and WZ backgrounds which are not accounted for in the present study. See for instance Fig. 2 of arxiv:1709.05822.

  • validity: high
  • significance: high
  • originality: high
  • clarity: good
  • formatting: excellent
  • grammar: excellent

Author:  Christopher W. Murphy  on 2019-11-24  [id 655]

(in reply to Report 2 on 2019-09-23)
Category:
reply to objection

Hello,

Thank you for the helpful feedback on my manuscript! I had originally based my analysis off of Ref. [45] (1510.01691), but I totally agree that it makes more sense to use Ref. [46] (1812.07591) as a benchmark as its set up is relevant for what will actually be measured at HL-LHC. The particular changes I made in responses to the points you raised are (1) I am now decaying the $W$ bosons and (2) the cut on $m_{jj}$ is now 850 GeV. The last paragraph in Section 3.4 is a comparison between Ref. [46] and my work. I find a larger AUC for $\Delta\phi_{jj}$ than Ref. [46], but this is likely due my approximation of using parton from the hard scattering process as jets. Other than that the results of the two works are generally consistent.

Best wishes,
Chris

---

## Round 4 · Referee Report · Anonymous (Referee 2) · 2019-11-24

Report

The manuscript was revised, addressing the points previously raised by the referee. I believe that the manuscript is ready for publication.

---

## Round 4 · Referee Report · Tilman Plehn (Referee 1) · 2019-11-28

Report

Thank you for considering all my comments. Let's publish!

---

## Round 4 · List of Changes

I have made the following changes based on the referee report of the anonymous referee: 1.) I am now decaying the $W$ bosons 2.) The threshold on $m_{jj}$ has been raised to 850 GeV

I have made the following changes based on the referee report of Tilman Plehn: 1.) I revised a paragraph in Section 2 discussing the precison-recall curve so that it goes into more detail. 2.) I added several paragraphs to Section 2 going into greater detail on focal loss and what it is useful for. In a nutshell it down weights easy-to-classify examples in the loss function so that training is more focused on the hard-to-classify examples. 3.) What was footnote 1 has been removed 4.) Feynman diagrams have been added, see Figure 1 5.) Figure 2 has been added, which gives both the precision-recall curve and the ROC curve for the models in Table 1. 6.) The cuts and feature engineering have been updated to match for Ref. [46] as closely as possible. Good agreement is found when a fair comparison can be made. I find the AUC to be higher for $\Delta\phi_{jj}$ than Ref. [46]. However this is due to my treatment of partons from the hard scattering process as jets. 7.) There is no problem with the signal-background separation in (what is now) Figure 3. The mean predicted probability for a classifier with an unweighted loss function trained on an imbalanced dataset is $r$, the imbalance ratio. I would take complete signal-background separation in the training dataset to be a sign of overfitting if such behavior was not also observed in the validation dataset, which it’s not in this case. Balancing the training set moves the mean value to 0.5. This can be seen in the upper right panel from the balanced random forest. Weighting the loss function with the inverse of the class frequencies also moves the mean value to 0.5. Focal loss is intermediate between these two scenarios, $r$ and 0.5, as can be seen in the bottom right panel. 8.) Performance is reported as the mean $\pm$ the standard deviation of the five folds of the cross validation. I added this to the caption of the table. It was already in the text. My apologies for the confusion around the use of the word significant. This is certainly not a Higgs boson level discovery, nor even 3-sigma evidence for something. Apparently I’ve been out of physics for too long. I was approaching from of the point of view how large of an effect would need to be observed to convince my business stakeholders it would be worth rolling this out. The manuscript has been revised accordingly. 9.) The global fit reference now includes citations for 9 papers, Ref. [56-64], and I added a footnote about 0904.3866. 10.) I added a paragraph to end of Section 4.1 attempting to clarify these questions. 11.) The precision-recall curves for the tests in Table 2 have been added in Figure 3. The curves are consistent with the finding in Table 2, and provide another way of demonstrating that class imbalance techniques can lead to a performance in charm tagging efficiency. 12.) A direct comparison against (what is now) Ref. [76] is not possible given the backgrounds considered in the two works are different.

Additional Notes: - 3 figures have been added - 10 references have been added

---

## Editorial Decision

published